# Determining the Economically Optimum Metaphylactic Strategy for Cattle Cohorts of Varied Demographic Characteristics

**DOI:** 10.3390/ani14101423

**Published:** 2024-05-10

**Authors:** Dannell J. Kopp, Robert L. Larson, Phillip A. Lancaster, Bradley J. White, Kristen J. Smith, Dustin L. Pendell

**Affiliations:** 1Beef Cattle Institute, Department of Clinical Sciences, College of Veterinary Medicine, Kansas State University, Manhattan, KS 66506, USA; 2Private Consultant, Baldwin City, KS 66006, USA; 3Department of Agricultural Economics, College of Agriculture, Kansas State University, Manhattan, KS 66506, USA

**Keywords:** bovine respiratory disease, economics, metaphylactic antibiotics

## Abstract

**Simple Summary:**

The administration of antibiotics to a whole cohort of high-risk feedlot cattle upon arrival to the feedlot is a common practice used by producers to control respiratory disease. A model was developed that utilizes an economic endpoint to determine the best application of antibiotics based on demographics of the cattle typically included in feedlot records. Determining the best situations to use and not use antibiotics helps to not only decrease cases of respiratory disease but also to ensure continued usefulness of antimicrobials. The goal of this study is to identify key characteristics of cohorts of cattle that did not receive metaphylaxis but that would benefit economically from the use of arrival antibiotics.

**Abstract:**

Metaphylactic antibiotic use in feeder cattle is a common practice to control respiratory disease. Antimicrobial stewardship is important to ensure continued efficacy and to protect animal welfare. The objective of this study is to identify characteristics of cohorts of cattle that had not received metaphylaxis that would have benefited economically from the use of metaphylaxis. Cohorts (*n* = 12,785; 2,206,338 head) from 13 feedlots that did not receive metaphylaxis were modeled using an economic model to estimate net returns for three metaphylactic options. Logistic regression models with covariates for entry weight, sex, average daily weight gain, number of animals per cohort, and days on feed, with feedlot as a random effect, were used to determine the model-adjusted probability of cohorts benefiting economically from metaphylaxis. Most (72%) cohorts in this data set that had not received metaphylaxis at arrival would not economically benefit from metaphylaxis. Sex, entry weight category, number of cattle in the cohort, and average daily weight gain were associated with the likelihood of benefitting economically from metaphylaxis. The results illustrated that cattle cohort demographics influenced the probability that cohorts would benefit economically from metaphylaxis and the type of metaphylaxis utilized, and integrating this information has the potential to influence the metaphylaxis decision.

## 1. Introduction

The bovine respiratory disease (BRD) complex is a multi-faceted disease involving bacterial and viral agents, environmental factors, commingling, transport, and other stressors [1,2,3,4,5]. In 2010, cattle death losses in the United States due to respiratory problems equated to $643,146,000 [6]. While mortality due to BRD may be a major economic loss, BRD morbidity is also important when including costs of treatment, labor, reduced average daily weight gain (ADG), and feed conversion [7,8,9,10,11]. Negative effects on ADG result in smaller, lower-marbling carcasses, with losses ranging from $23.23 to $54.01 per carcass for cattle treated for BRD [12]. Cattle treated for BRD more than once also result in lower carcass grades compared to those that were only treated once or not at all [13]. Combining all sources of potential loss, BRD leads to substantial economic losses for cattle producers. Decreased prevalence of BRD within the livestock industry can also lead to varied effects on many sectors of the industry, including potentially lowering beef prices for the consumer and benefiting society [14]. A true evaluation of BRD cost should include mortality and all the expenses associated with morbidity [15].

Antimicrobial metaphylaxis has been implemented in an effort to curb economic losses, as well as improve the animal welfare of cattle entering feedlots [16]. Metaphylaxis is the use of a U.S. Food and Drug Administration (FDA)-approved antimicrobial agent to treat an entire cohort of cattle that are at high risk for a disease process that would be modified by treatment at or near the time of arrival to a feedlot [17]. Decreased risk of BRD-related morbidity and mortality are commonly seen results of implementing metaphylaxis [18,19,20,21].

The impact of BRD is highly variable among cohorts of cattle [22,23]. In addition, as an indication of metaphylactic efficacy, odds ratios for BRD morbidity cumulative incidence and BRD mortality cumulative incidence vary greatly between different comparisons of common metaphylactic drugs [24]. The combination of inconsistent disease impact and variable metaphylactic efficacy makes it difficult to predict which cohorts will benefit from metaphylactic treatment for the control of BRD.

Currently, metaphylaxis decisions are made based on risk classification of the cohort [17,19]. Risk classification assigned at the feedlot is made based on both subjective and objective measures. While much work has been done to show that metaphylaxis can reduce morbidity and mortality, because of differences in efficacy and cost between metaphylactic options, optimum metaphylaxis decisions must be based on more information than risk classification alone [25,26]. Looking at the decision strictly from an economic outcome helps remove some subjectivity, and incorporating metaphylactic efficacy and cost brings additional important factors to the decision-making process.

The objective of this study is to identify characteristics of cohorts of cattle that had not received metaphylaxis that would have benefited economically from the use of metaphylaxis. Specifically, three metaphylactic options are evaluated: no metaphylaxis (NOMET), low-cost/low-efficacy metaphylaxis (LCLE), and high-cost/high-efficacy metaphylaxis (HCHE), to discover information to update antimicrobial use decisions leading to improved antimicrobial stewardship and economic success. The development of an economic model to determine the differences in net returns between the three metaphylactic options will be used to identify cohort demographics associated with each optimized metaphylactic option.

## 2. Materials and Methods

### 2.1. Ethical Statement

Institutional Animal Care and Use Committee (IACUC) approval was not required as historical operational data were used for the analysis and no procedures were performed on cattle specifically for this research.

### 2.2. Study Design

Retrospective, observational, cohort-level data from 13 feedlots over a 5-year period (2016–2020) were collected under confidentiality agreements with feedlot collaborators. Data were combined with economic and statistical models to assess the potential economic impact of metaphylaxis in cohorts that did not receive metaphylaxis. A cohort was defined as a purchased group of cattle that entered and exited the feedlot together. BRD morbidity was defined as cattle identified with BRD and treated with antimicrobials. BRD mortality was defined as cattle that died following treatment for BRD or that were classified as dying of BRD even if not previously treated (i.e., pen deads).

### 2.3. Exclusion Criteria

Exclusion criteria were applied to the feedlot data to remove potential data entry errors and to guarantee that only cohorts with complete data were included. For this analysis, only cohorts that did not receive metaphylaxis upon arrival were included. The average entry weight of cohorts had to be greater than 181 kg and less than 499 kg. Cohorts of mixed sex were excluded, in order to be able to analyze differences in economic benefits between cohorts of steers and cohorts of heifers. Data were filtered to only include records where cohort size was greater than 40 cattle at arrival. Total cohort days on feed was limited to less than 400 days and greater than 70 days. Cohort BRD morbidity risk was required to be greater than or equal to BRD mortality risk, meaning the case fatality risk of the cohorts could not be greater than 1, which would have allowed for more BRD deaths than calves treated for BRD.

### 2.4. Study Variables from Cohort Records

The following data were acquired from cohort records: number of head purchased, whole-cohort entry weight, number treated for BRD, number of deaths attributed to BRD, days on feed, number of head sold, whole-cohort exit weight, and pounds of feed consumed. The subsequent adjusted and calculated variables were computed in order to have a consistent comparison between metaphylactic options. All cohorts were adjusted to a constant entry date of 5 October 2018, with exit dates based on actual days on feed (constant entry date allowed for the comparison of cohort demographic effects on the economic outcomes while holding cattle and feed prices steady). The whole-cohort entry weight was divided by number of head purchased to determine the average entry weight per head.

The average daily weight gain of cattle within a cohort was calculated as the average entry weight per head subtracted from the average exit weight per head divided by days on feed to compute deads-out ADG. Bovine respiratory disease morbidity and mortality risk was provided in the original data from collaborating feedlots. Morbidity attributed to other causes beyond BRD was not included in the model. The case fatality risk as a depiction of treatment success was calculated as number of deaths from BRD divided by number of cattle treated for BRD within each cohort. Deaths not attributed to BRD were designated as “other deaths” and were assumed to be equal proportions regardless of metaphylactic options.

Head days for each cohort were calculated by assuming that BRD deaths occurred at day 50, per expert opinion. Other deaths were assumed to occur at the midpoint for days on feed for that cohort, and cattle that finished the feeding phase had head days equivalent to days on feed. Head days were used to calculate expenses accrued on a daily time-step by the number of cattle in the cohort present at each day, such as yardage and feed cost.

Arrival processing and yardage costs were acquired for each cohort from the cooperating feedlot. The whole-cohort arrival processing cost was divided by the number of cattle upon feedlot arrival and then averaged across all cohorts to determine an average processing cost per head across all yards. The processing cost was assigned as $23.60 per head upon feedlot entry and was based on data provided by cooperating feedlots. The whole-cohort yardage cost was divided by head days and then averaged across all cohorts to determine an average yardage cost per head day across all yards. A yardage cost of $0.35 per head day was used in the model.

### 2.5. Study Variables—Cattle Prices

Feeder cattle purchase prices for 5 October 2018 (Table 1) [27,28,29,30,31] were acquired from an online database. A purchase cost was then calculated based on the average entry weight for each cohort and the purchase price relative to entry weight category. Live cattle sale prices were the weekly prices reported from November 2018 to October 2019 (Table 2) [32]. The sale price was assigned based on the cohort exit date from the feedlot, with the exit date being equivalent to 5 October 2018, plus the actual days on feed of the cohort. The sale price and the cohort exit weights were used to determine the income for each original cohort and its two metaphylaxis treatment-generated cohorts.

### 2.6. Study Variables—Metaphylactic Treatment Prices

The pricing of metaphylactic options was based on various antibiotic agents at bulk price from an online veterinary retailer as of June 2022, in combination with expert opinion [33]. The costs for the LCLE and HCHE metaphylactic options were set at $0.02 per kg and $0.08 per kg of entry weight, respectively. The cost of metaphylaxis was in addition to the processing costs.

### 2.7. Study Variables—Feed Costs

The feed ration price per kg was determined using historical commodity and feedstuffs prices. Kansas State University’s Focus on Feedlots Monthly Report was used for alfalfa hay and corn prices [34]. Agriculture Marketing Resource Center’s Weekly Ethanol, Distillers Grain, and Corn Prices was used to determine the price of wet distillers grains based on an average of the monthly prices from Illinois, Iowa, Nebraska, South Dakota, and Wisconsin [35]. A feed ration consisting of 10% alfalfa hay, 60% corn, and 30% wet distillers grains was developed. Prices for alfalfa, corn, and wet distillers grains were determined to be $170.24 per ton ($0.19 per kg), $137.50 per ton ($0.15 per kg), and $46.12 per ton ($0.05 per kg), respectively. Using these values, feed was priced at $0.18 per kg of feed. For NOMET, the total pounds of feed consumed was acquired from data records and used to determine feed costs. When cohort data were modified for LCLE and HCHE metaphylaxis treatment-generated cohorts, the total pounds of feed consumed was modified by a proportion of change in head days to account for increased head days within the cohort due to fewer BRD deaths.

### 2.8. Study Variables Based on Differences in Metaphylactic Efficacy on Odds Ratios for Cumulative BRD Morbidity

A systematic review and mixed treatment meta-analysis of metaphylaxis was used to identify that there are differences in efficacy among metaphylactic antimicrobials which allowed us to create higher-efficacy and lower-efficacy categories [24]. These higher- and lower-efficacy categories were not intended to indicate specific antimicrobials but were used to inform the metaphylactic efficacy odds ratios of 0.6 and 0.3 for LCLE and HCHE, respectively.

Each cohort was modeled three times: first, the actual performance of the cohort (i.e., no-metaphylaxis NOMET); second, the modeled performance of the cohort if it had received LCLE; and third, the modeled performance of the cohort if it had received HCHE. The metaphylactic efficacy odds ratio was used to modify the morbidity risk of each NOMET cohort to determine a new, reduced morbidity risk within the cohort when either LCLE or HCHE metaphylaxis was modeled. The new morbidity risk was used to determine the number of animals within each metaphylaxis treatment-generated cohort that was treated for BRD. The original case fatality risk of each baseline cohort (NOMET) was then used to determine the number of animals that died from BRD for each metaphylaxis treatment-generated comparison cohort. Other deaths were assumed to not be differentially affected by the use of a metaphylactic antibiotic. A lower morbidity risk and subsequently lower mortality risk increased both income and costs for LCLE- and HCHE-modeled cohorts compared to the income and costs modeled for the associated NOMET cohort.

Using each NOMET cohort’s ADG we calculated a healthy-adjusted ADG, as well as a BRD-treated-adjusted ADG, in relation to the number of morbid calves in the NOMET cohort. These were calculated on the basis of a 5% decrease in weight gain of the BRD-treated calves [12]. These adjusted ADG values were used to determine the group-level ADG for each metaphylaxis treatment-generated cohort based on the adjusted number of healthy and BRD-morbid cattle for both LCLE and HCHE options. The group-level ADG was then used to calculate the total exit weight for each metaphylaxis treatment-generated cohort.

The number of head days for the LCLE and HCHE cohort models were based on the metaphylaxis treatment-adjusted number of cattle that died from BRD, relative to the designated morbidity and mortality reduction assigned for the LCLE and HCHE, and the number of other deaths provided in the NOMET cohort. Bovine respiratory disease deaths were assumed to die on day 50 while other deaths were assumed to die at the midpoint of days on feed.

Morbid cattle that were pulled and treated for BRD were charged for the high-cost antibiotic priced at $0.08 per kg as well as being charged a $1.50 chute fee per head. For dosing purposes, the weight of morbid cattle was based on the average cohort entry weight into the feedlot.

### 2.9. Model Comparison of Net Return between Metaphylactic Options

A net return economic model using enterprise budgets was developed to assess the optimal metaphylactic option of cohorts of U.S. feedlot cattle that had not received metaphylaxis at arrival based on the highest net return among the three metaphylactic options: NOMET, LCLE, and HCHE. For analysis, a logistic regression model was used to initially compare NOMET to a “yes” metaphylaxis variable (YESMET) which comprised both LCLE and HCHE, followed by a second logistic regression model to analyze LCLE versus HCHE within the YESMET-modeled cohorts.

Net returns were calculated as the difference between total revenue and total expenses. Total expenses included the purchase cost of feeder cattle, processing cost, yardage cost, BRD treatment cost, chute fee cost, and feed cost, as well as the cost of metaphylaxis. Net returns for all cohorts were compared across NOMET, LCLE, and HCHE, and the optimal decision for each cohort was selected based on the highest net return.

Economic calculations and metaphylaxis adjustments were completed using Excel (Microsoft Excel, Microsoft Office Professional Plus 2019, Microsoft Corporation, Redmond, WA, USA). Data analysis was completed using RStudio (R Core Team, 2022). The association of sex, entry weight, cohort size, exit weight, ADG, and days on feed with the probability of benefiting economically from metaphylaxis options was evaluated with a logistic regression model with feedlot as a random effect using the glm() function. Data variables were evaluated for interactions and none were found. Two logistic regression models were completed with the first model comparing NOMET to a “yes” metaphylaxis variable that combined LCLE and HCHE into one outcome, and the second model comparing LCLE to HCHE within the cohorts that were YESMET in the first regression analysis. Among cohorts benefiting economically from metaphylaxis, the association of the same variables with the probability of benefiting from HCHE metaphylaxis versus LCLE metaphylaxis was evaluated with a logistic regression model with feedlot as a random effect.

## 3. Results

Records for 16,809 cohorts were originally acquired from collaborators. The application of the exclusion criteria outlined above resulted in 12,785 cohorts of feedlot cattle in the dataset. The final dataset of 12,785 cohorts encompassed 4038 cohorts of heifers and 8747 cohorts of steers. These cohorts included 2,206,338 head of cattle that originated from 38 states across the United States; 631,521 head were heifers and 1,574,817 head were steers. The analysis of the included cohorts found the median percent morbidity risk was 5.1% with an interquartile range of 2.5% to 10% (Figure 1).

Metaphylaxis adjustments (LCLE and HCHE) resulted in a greater sale weight and less BRD treatment costs, as well as an increased arrival processing cost, increased feed cost due to increased feed consumed, increased yardage, and other costs associated with less disease and death compared to NOMET. The optimal metaphylactic choice based on the economic model output varied among cohorts with a distribution of cohorts falling into NOMET (72.0%; 9204/12,785), LCLE (25.4%; 3242/12,785), and HCHE (2.7%; 339/12,785). Statistical analysis included two logistic regressions that first compared NOMET (72%) to YESMET (28%) and then compared LCLE to HCHE within the YESMET cohorts.

When comparing NOMET to YESMET, sex, the entry weight category, number of cattle in the cohort, and the average daily weight gain category were found to be statistically significant. Cohorts of steers had a higher probability (31.2 ± 3.3%) of benefiting economically from metaphylaxis in comparison to cohorts of heifers (19.9 ± 2.5%) (*p* < 0.05). Cohorts within the three lowest average entry weight intervals (37 kg intervals) had a higher probability of benefiting economically from metaphylaxis than the other four weight categories (Figure 2). Cohorts of less than 100 head of cattle had a higher probability (30.0 ± 3.3%) of benefiting economically from metaphylaxis in comparison to cohorts with 100 to 200 head of cattle (25.1 ± 2.9%) (*p* < 0.05) and those cohorts with over 200 head of cattle (20.8 ± 2.6%) (*p* < 0.05). Cohorts with an ADG of 0.45–1.36 kg/day had a higher probability (8.3 ± 0.5%) of benefiting economically from metaphylaxis compared to those with ADG greater than 1.36 kg/day (Figure 3).

Within the YESMET cohorts, the entry weight category, number of cattle in the cohort, and average daily weight gain were associated with the probability of HCHE versus LCLE metaphylaxis having a higher net return. Of cohorts benefiting economically from metaphylaxis, those within the three lowest entry weight categories had the highest probability of benefiting economically from HCHE metaphylaxis (Figure 4). Of cohorts that benefited from metaphylaxis, cohorts of less than 100 head had a higher probability (5.3 ± 1.4%) (*p* < 0.05) of benefiting from HCHE metaphylaxis in comparison to cohorts of 100 to 200 head of cattle (3.3 ± 0.9%) and cohorts of over 200 head of cattle (3.2 ± 0.9%). Of those that benefited from metaphylaxis, cohorts with an ADG of 0.45–1.36 kg/day had the highest probability of benefiting economically from HCHE metaphylaxis and cohorts with an ADG of 1.36–1.54 kg/day and 1.54–1.68 kg/day had a higher probability of benefiting economically from HCHE metaphylaxis compared to cohorts with an ADG of 1.68–2.72 kg/day (Figure 5).

## 4. Discussion

Modeling the administration of metaphylaxis changed both income and costs resulting in different net returns for the three modeled scenarios for each original cohort. When comparing cohorts of feedlot cattle that had not received metaphylaxis across NOMET, LCLE, and HCHE, most (72%) cohorts in this data set would not economically benefit from metaphylaxis. Our results differ from Dennis et al. [26] who modeled net return distributions for metaphylactic decisions for cattle at high risk for BRD. Because the data set used in our study only included cohorts that did not receive metaphylaxis, we assume that the perceived risk of BRD in our cohorts was lower than for cohorts modeled by Dennis et al. [26].

Based on the cohorts used in this analysis, we found a median cohort percent morbidity of 5.1%. Prior research showed a 12.8% bovine respiratory disease morbidity risk in feeder cattle via surveys of feedlots in Kansas, Nebraska, Texas, North Dakota, and other states, which differs substantially from our results [36]. Because we excluded cohorts that received metaphylaxis at arrival, it is not surprising that morbidity was lower in our study. Cattle chosen to receive metaphylaxis at feedlots typically have an increased risk of developing disease, so exclusion of these animals would lead to decreased morbidity within this data set [17].

Steers had a greater probability of benefiting economically from metaphylaxis compared to heifers. Other studies have estimated that steers have a mean ADG of 1.62 kg per day while heifers have a mean ADG of 1.36 kg per day [37]. With this in mind, a 5% decrease in ADG seen in cattle affected by BRD would have a greater impact on cohorts of steers because of their greater ADG. Metaphylaxis thus provides more improvement in sale weight within steer cohorts over heifer cohorts.

Upon arrival to the feedlot, lightweight cohorts of cattle are considered to be at greater risk for BRD in comparison to heavier cattle [17]. The same BRD odds ratio benefit from metaphylaxis applied to high-morbidity-risk cohorts provides a greater reduction in number of cattle affected by BRD compared to low-morbidity-risk cohorts; therefore, metaphylaxis, and specifically, HCHE metaphylaxis, is more likely to provide economic benefits for high-risk cohorts.

Cohorts with less than 100 head of cattle were more likely to benefit economically from metaphylaxis. We do not have a hypothesis for the biological association between smaller cohort size and the economic benefit of metaphylaxis. A smaller cohort size would be associated with a proportionally higher impact of morbidity and mortality. One morbid animal in a pen of 50 head (1/50, 2.0%) would cause a significantly higher morbidity compared to a single morbid animal in a pen of 500 head (1/500, 0.2%). Another possibility for this association is that smaller cohorts of animals may be more likely to be commingled in order to fill a pen. Commingling means that cattle from different sources are put together in order to fill a pen closer to capacity. The commingling of cattle from different sources increases the likelihood of cattle developing BRD [38]. Each smaller cohort will have different levels of immunity to diseases as well as bring in different strains of bacteria and viruses, which helps to contribute to an increased likelihood of developing disease. We were surprised to see that cohorts with lower ADG were more likely to benefit from metaphylaxis and HCHE metaphylaxis.

The limitations within this study include the following: the choice of constant entry dates and pricing; set values for feed costs, chute fees, yardage costs, processing costs, and metaphylactic prices and efficacies. Constant entry dates and pricing allow for a more straight-forward comparison of net returns based solely on metaphylaxis effects. The date of 5 October 2018 was chosen to allow all cohorts to reach slaughter prior to the COVID-19 pandemic, as there were drastic changes in market values during those times. We discern that these values would be a better representation of actual figures if they were allowed to fluctuate, but the additional variation would hinder the comparison of cohort demographics across equivalent economic factors of production. Future research utilizing stochastic prices for cattle purchasing, sales, and various inputs would be useful for determining how prices affect metaphylactic decision making.

## 5. Conclusions

By modeling cohorts of feeder cattle for NOMET, LCLE, and HCHE metaphylaxis options, we were able to identify which strategy resulted in the greatest net return, and we were able to analyze key differences and similarities between the populations. We found that most (72%) cohorts in this data set that had not received metaphylaxis at arrival would not economically benefit from metaphylaxis, and we determined that cattle cohort demographics influenced the probability that cohorts would benefit economically from metaphylaxis and the type of metaphylaxis utilized. The results of this study as well as future published studies will provide decision makers with important information to improve the integration of multiple considerations when making metaphylaxis decisions in cohorts of cattle and management systems similar to those modeled in this study. Based on the results of this study, decision makers should recognize that NOMET was the correct decision in 72% of comparisons, and they should consider using metaphylaxis in cohorts currently not receiving the intervention when the arrival weight is low, the cohort consists of a small number of cattle, and the cohort sex is steers. In order to optimize net returns while maintaining antimicrobial stewardship and animal welfare, producers and veterinarians should continue to develop decision tools that incorporate cohort demographic factors when making metaphylaxis choices.

## Figures and Tables

**Figure 1 animals-14-01423-f001:**
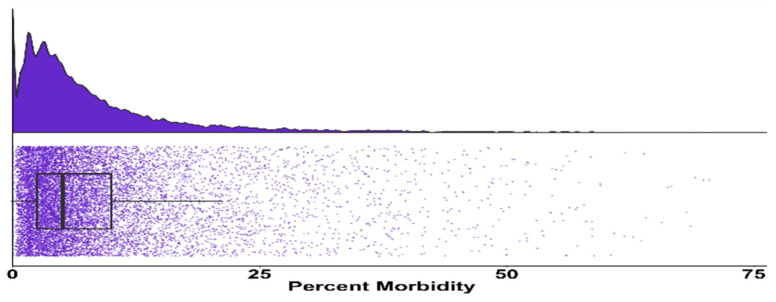
Distribution of cohort morbidity risk displayed in a raincloud plot. The upper section shows density of morbidity percentage due to bovine respiratory disease for each cohort. The lower portion shows a scatter plot of each cohort by morbidity percentage with a box-and-whisker plot overlay.

**Figure 2 animals-14-01423-f002:**
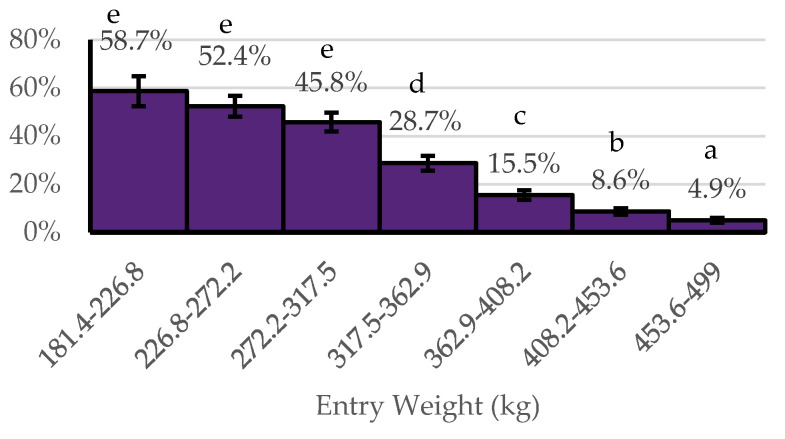
Probability of benefiting economically from metaphylaxis (YESMET) by entry weight. ^abcde^ Bars without a common superscript letter differ at *p* ≤ 0.05.

**Figure 3 animals-14-01423-f003:**
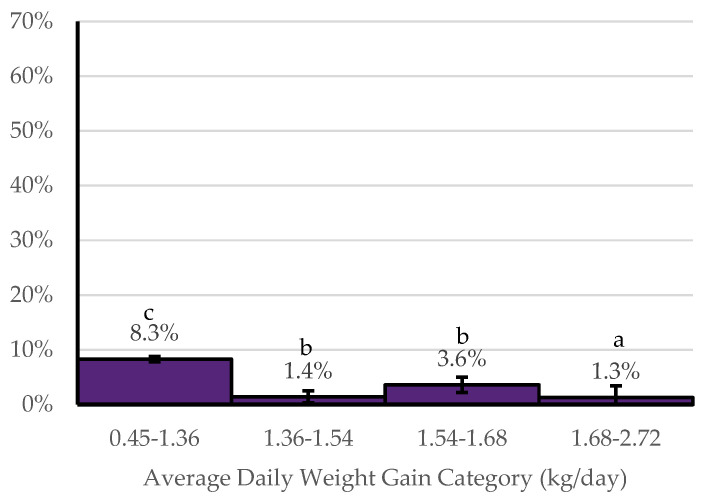
Probability of benefiting economically from metaphylaxis (YESMET) by average daily weight gain categories. ^abc^ Bars without a common superscript letter differ at *p* ≤ 0.05.

**Figure 4 animals-14-01423-f004:**
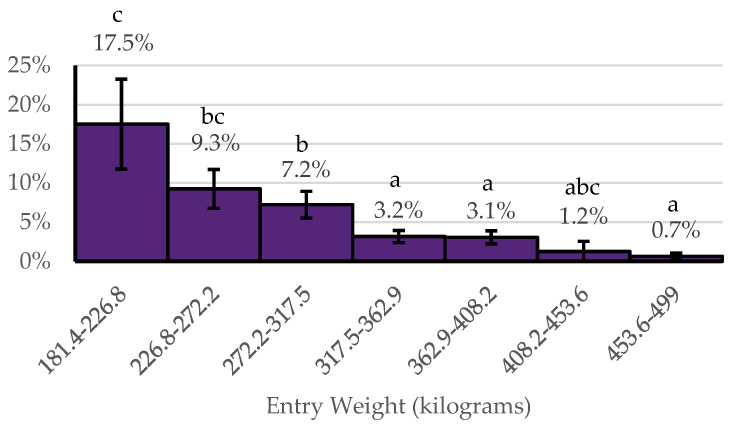
Among cohorts benefiting economically from metaphylaxis (YESMET), the probability of benefiting economically from HCHE by entry weight. ^abc^ Bars without a common superscript letter differ at *p* ≤ 0.05.

**Figure 5 animals-14-01423-f005:**
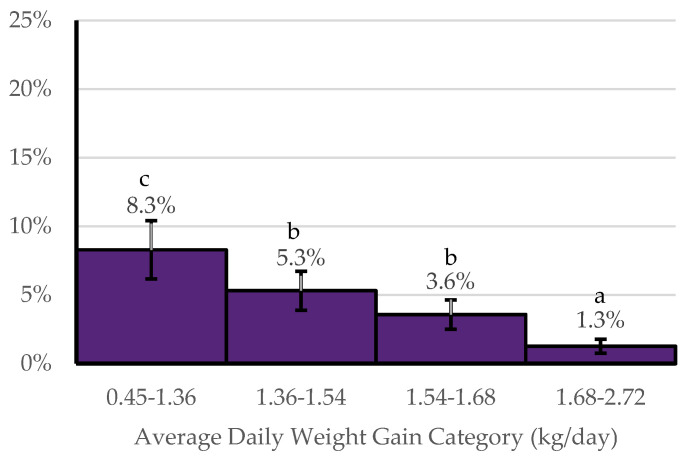
Among cohorts benefiting economically from metaphylaxis (YESMET), the probability of benefiting economically from HCHE by average daily weight gain categories. ^abc^ Bars without a common superscript letter differ at *p* ≤ 0.05.

**Table 1 animals-14-01423-t001:** Feeder cattle purchase prices as of 5 October 2018 [27,28,29,30,31].

Entry Weight Category	Price (USD/kg)	Price (USD/cwt)
181.4–226.8 kg:	4.11	186.36
226.8–272.2 kg:	3.83	173.65
272.2–317.5 kg:	3.60	163.28
317.5–362.9 kg:	3.50	158.73
362.9–499.0 kg:	3.39	153.67

**Table 2 animals-14-01423-t002:** Fed cattle prices, monthly range of weekly sale prices [32].

Sale Month	Price (USD/kg)	Price (USD/cwt)
November 2018:	2.5–2.59	113.53–117.54
December 2018:	2.61–2.7	118.40–122.65
January 2019:	2.7–2.74	122.63–124.06
February 2019:	2.73–2.78	123.73–126.15
March 2019:	2.77–2.83	125.83–128.50
April 2019:	2.76–2.82	125.19–127.75
May 2019:	2.54–2.92	115.28–132.27
June 2019:	2.43–2.5	110.29–113.48
July 2019:	2.43–2.5	110.41–113.37
August 2019:	2.31–2.49	104.89–112.78
September 2019:	2.2–2.31	99.86–104.61
October 2019:	2.34–2.43	106.26–110.03

## Data Availability

Data were provided through agreements with collaborating entities and data are bound by confidentiality agreements; thus, raw data cannot be shared.

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
