# Peer review of "Determining the Economically Optimum Metaphylactic Strategy for Cattle Cohorts of Varied Demographic Characteristics"

_animals, 2024, doi:10.3390/ani14101423_

Round 1

Reviewer 1 Report

Comments and Suggestions for Authors

Dear Author , Thank you for submitting the manuscript. The manuscript entitled "Determining the economically optimum metaphylactic strategy for cattle cohorts of varied demographic characteristics" is very unique and novel. Certainly, it is require to make a model that can let you the selection of different antibiotics, other spending and other stuff based on economically derived different models. 

Another important ascpect is the antimicrobial stewardship. By using this it is certain that unnecessary use of antimicrobial can be reduce and farmers can get benefit of it.

Please address following:

1) how can you apply it to the field or how can anyone can utilise it? have you tried to implement through a software or any simple programming ?

2) have you think of incorporating AI tool to train the model based on the data as a future prospects?

3) is it geographically restricted or any where at US, it can be used by a farmer/producers? can dataset and model support usage ?

4) what other parameters can be looked ?

Comments on the Quality of English Language

minor

Author Response

To the reviewers for manuscript 2963180:

We thank you for your thoughtful and helpful reviews. We addressed your comments and agree that your suggestions and questions helped to improve the clarity of the manuscript.

Sincerely,

Dannell J. Kopp, Robert L. Larson, Phillip A. Lancaster, Bradley J. White, Kristen J. Smith, and Dustin L. Pendell

Reviewer 2 Report

Comments and Suggestions for Authors

The manuscript was well-written and easy to read. However, there are areas that require clarification or could be expanded upon. 

See attached. 

Author Response

(The authors gave the same response as above.)

Reviewer 3 Report

Comments and Suggestions for Authors

This is a theoretical study based on the modeling of the possible economic effects of different options of metaphylaxis in feedlots. Although the work is interesting, the writing must be improved to be published. The material and methods must be clarified and detailed with higher precision. In addition, some essential results as univariate and multivariate analysis must be shown. The discussion should be deeper. Finally, some references must be updated, most of the links in the references are not working and some references, as reports or proceedings, are not properly cited in the reference list.

More specific, but not exhaustive, comments are now detailed:

The most striking results would be that “Most (72%) cohorts did not economically benefit from metaphylaxis”. This results statement is included in the abstract but is not clearly shown in the results and neither discussed. This should be shown in the results and discussed.

Line 39, In addition to commingling, transport is usually highlighted as factors involved in BRD. References 2 and 3 about BRD etiology should be adapted. I suggest these two references: 1) Griffin, D.; Chengappa, M.M.; Kuszak, J.; McVey, D.S. Bacterial pathogens of the bovine respiratory disease complex. Vet. Clin. Food Anim. 2010, 26, 381–394. And 2) Taylor, J. D., Fulton, R. W., Lehenbauer, T. W., Step, D. L., & Confer, A. W. (2010a). The epidemiology of bovine respiratory disease: What is the evidence for predisposing factors? The Canadian Veterinary Journal, 51(10), 1095. /pmc/articles/PMC2942046/

Lines 52-58 Metaphylaxis is also used in Canadian feedlots and other countries with beef industry, including European countries and Australia. It would be interesting to include this information on metaphylaxis also from this more general point of view.

Lines 59-64. The statement “Impact of BRD is highly variable among cohorts of cattle” needs a reference. Efficacy of metaphylactic is mentioned but is not defined. Odds ratios are mentioned, but what are those odds ratios comparing (morbidity, mortality, both, daily weight gains…? Please define metaphylactic efficacy in the context of this study.

Lines 65-72. A reference is needed for “Currently, metaphylaxis decisions are made based on risk classification of the cohort.” The statement “While much work has been done to show that metaphylaxis can reduce morbidity and mortality, because of differences in efficacy and cost between, and variation within, metaphylactic options, optimum metaphylaxis decisions must be based on more information than risk classification alone” seems an opinion. Please, include references of much of that work done in effects of metaphylaxis reducing morbidity and mortality and about differences in efficacy and cost.

Lines 73-83. The three metaphylactic options evaluated (no metaphylaxis (NOMET), low-cost/low-efficacy metaphylaxis (LCLE), and high-cost/high-efficacy metaphylaxis (HCHE)) are mentioned for the first time.  It is essential to clearly define these three options that will be compared. In my opinion, they will be defined in material and methods. In addition, the explanation that “For analysis, NOMET was initially compared to a “yes” metaphylaxis variable (YESMET) which was comprised of both LCLE and HCHE, followed by an analysis of LCLE versus HCHE within the YESMET modeled cohorts” will be better included in material and methods.

In material and methods make sub-sections to make it easier to follow. For instance, Ethic statement, study design or cohorts, inclusion (or exclusion) criteria, variable studied in each cohort, estimated variables at cohort level (entry date, average entry weight, average daily wight gain, morbidity and mortality risk, case fatality risk, head-days), creation of the cohorts with different metaphylaxis options, costs and net returns, …

Lines 131-142. “creation” of the two metaphylaxis-treatment cohorts (LCLE and HCHE) for each of the original NOMET cohorts must be clearly explained. Also, the meaning and the election of the odds ratios of 0.6 and 0.3. I do not understand why the cohorts are named Low cost/low efficacy and high cost/high, that is according both cost and efficacy, if only odds ratios of reduction of BRD are indicated (lines132-135). Also, how many cohorts in each category are finally comparing?

Line 140. which is the baseline cohort, is the NOMET cohort?

Lines 171-172. This information is table 1. It is redundant.

line 181. Table 1. USD/cwt. What is the abbreviation cwt for?

Line 213. The version and supplier must be included in the computer programs used

Line 233. Figure 1 is too simple. Complete with exclusion criteria or delete it.

Lines 238- 245. Where are these results shown, including the statistical analysis to show the significant differences?

Lines 246-49 Where are these results shown, including the statistical analysis to show that the differences are significant?

Author Response

(The authors gave the same response as above.)

Round 2

Reviewer 2 Report

Comments and Suggestions for Authors

I have no further questions or concerns.  All of my initial concerns have been addressed.

Reviewer 3 Report

Comments and Suggestions for Authors

thank you for accepting the suggested changes